# IMPROVE LLM PRE-TRAINING WITH RL-GUIDED AN-NEALING

## ABSTRACT

Training large language models (LLMs) typically proceeds in two distinct stages: pre-training and post-training. However, the question of how to exploit these stages synergistically—particularly how post-trained models can inform and improve pre-training—remains underexplored. We begin by analyzing training dynamics and identify the annealing (mid-training) phase as a critical turning point for the pre-trained base model's capabilities. During this stage, high-quality corpora are introduced under a rapidly decaying learning rate, leading to a substantial shift in the base model's probability distribution and a noticeable surge in performance. Interestingly, while reinforcement learning (RL) during post-training induces only minor distributional shifts, it significantly enhances reasoning capabilities. Motivated by this observation, we propose **RL-Guided Annealing (RGA)**, a method designed to leverage RL-enhanced models, naturally produced during standard LLM training pipeline, to guide token weighting during the annealing phase. Specifically, RGA transfers knowledge from the RL stage back to annealing by reassigning token-level importance weights based on the per-token loss differences between the base and RL models. Notably, RGA does *not* require any specially trained teacher or reference model. Across multiple model families, RGA consistently improves performance, achieving average gains of 5.21%, 1.84%, and 1.78% on 10 pre-training benchmarks. It also *boosts downstream performance after post-training* by over 2%. These findings reveal a positive feedback loop between pre-training and post-training: RL-tuned models retroactively improve their foundational base models, which in turn support more effective RL—enabling a self-reinforcing path toward higher model quality.

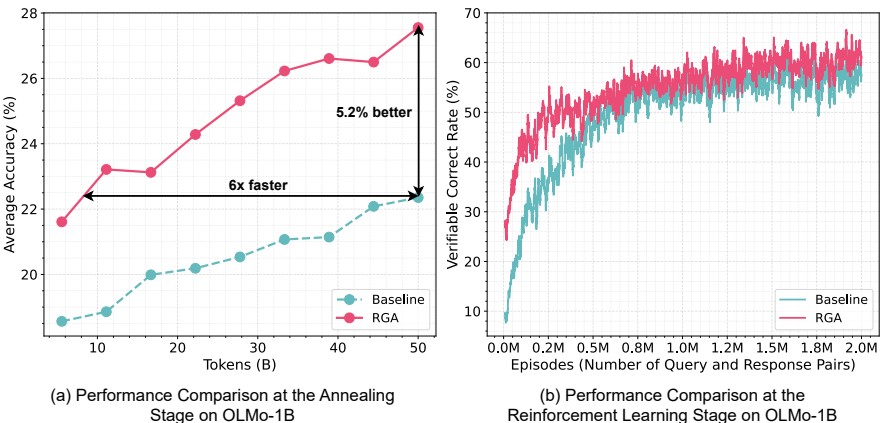

(a) Performance Comparison at the Annealing Stage on OLMo-1B

(b) Performance Comparison at the Reinforcement Learning Stage on OLMo-1B

Figure 1: RL-Guided Annealing (RGA) on OLMo-1B substantially outperforms baselines trained with standard annealing method. (a) At the annealing stage, RGA improves average accuracy across 10 widely-used benchmark tasks by 5.2% and reaches the baseline performance $6\times$ faster. (b) The improvements can carry over to post-training: during reinforcement learning (RL), RGA maintains a higher verifiable correct rate than the baseline and achieves better performance.

# 1 INTRODUCTION

Training large language models (LLMs) is a multi-stage process comprising pre-training and post-training (Achiam et al., 2023; Touvron et al., 2023; Yang et al., 2025a). Pre-training of LLMs typically optimizes a next-token prediction objective that applies a uniform weight to every token in the loss function. In contrast, reinforcement learning (RL) follows a different training paradigm: RL methods assign non-uniform token weights based on reward signals from the environment (Schulman et al., 2017; Shao et al., 2024), yielding substantial improvements in the reasoning capabilities. A strong pre-trained base model serves as the foundation for effective and efficient post-training. However, little research examines how to leverage these two stages synergistically, specifically, how post-trained models can provide guidance for pre-training, thereby improving final performance.

Much of the literature on improving pre-training focuses on data selection, which can be viewed as document-level reweighting. Methods in this line of work (Wenzek et al., 2019; Xie et al., 2023a; Albalak et al., 2024) primarily perform data cleaning to quickly filter noisy documents. Other studies investigate sample-level reweighting (Gu et al., 2023; Zhang et al., 2025), but these approaches often rely on heuristic rules and remain relatively coarse-grained. Recent research suggests that only a small subset of tokens is pivotal for model capability (Arbuzov et al., 2025; Cui et al., 2025), indicating that sentence-level upsampling or downsampling may be suboptimal and motivating finer-grained token-level reweighting.

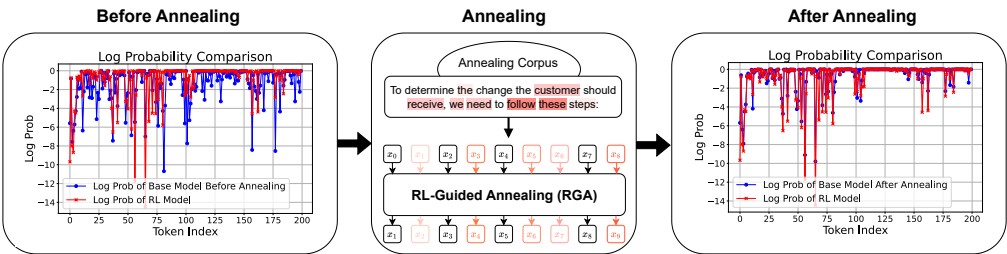

Figure 2: RGA enhances the annealing stage by dynamically reweighting tokens in the annealing corpus, thereby shifting the model's token distribution toward that of a more capable RL model.

By analyzing training dynamics during the pre-training phase, we identify the annealing (i.e., mid-training) stage as a critical turning point. This stage typically trains on the highest-quality corpora under a rapidly decaying learning rate to accelerate the acquisition of high-order capabilities such as graduate-level STEM knowledge or mathematical/code reasoning (Touvron et al., 2023; Yang et al., 2025a). Our analysis indicates that annealing induces a significant shift in the probability distribution of the base model. As shown in Figure 2, we compare the token-level log probability distributions over sequences produced by the final RL model with those of the base model before and after annealing. It can be observed that the RL model is much closer to the annealed model, while it differs significantly from the pre-annealed model. The observation suggests that annealing plays a pivotal role in the qualitative transformation of the base model. Since the RL model and the post-annealed base model exhibit similar token-level probability distributions, it is natural to consider whether the RL model can be used to refine the token loss distribution during annealing. By doing so, we aim to produce a model that is more closely aligned with the RL model, thereby yielding a stronger base. This improved base could be more compatible with the subsequent RL phase and ultimately lead to better overall performance.

Building on these observations, we propose RL-Guided Annealing (RGA). During the annealing stage, we use the RL-tuned model as a reference model to assign dynamic token-level weights. RGA leverages the RL model produced by the training pipeline itself, without requiring a separately trained teacher model on specially selected pretraining data as in Lin et al. (2024). RGA computes the delta values of per-token loss between the base model and the RL model and, by contextualizing these delta values with their sequence-wise distribution, derives relative importance weights for each token. Rather than discarding tokens via hard selection, we dynamically modulate their weights, thereby preserving the semantic coherence of the training data. Consequently, the pre-training stage

Figure 3: Token-distribution comparison between the pre-annealing base model and the RL model. Background intensity reflects $\Delta \log p = \log p_{\mathrm{RL}} - \log p_{\mathrm{base}}$, with deeper red denoting tokens on which the RL model is substantially more accurate.

effectively mimics the RL training paradigm by prioritizing tokens that contribute most to downstream performance, while largely retaining the efficiency of standard next-token prediction.

The core contribution of RGA lies in introducing a novel training framework—the first to leverage an RL-tuned model to guide **token-level weight assignment during the annealing phase** of pre-training. This paradigm establishes a **bidirectional synergy between pre-training and post-training** by reusing models already produced in the standard training pipeline. Notably, RGA achieves this without relying on any specially trained reference models or larger teacher architectures, making it both practical and scalable. Extensive experiments show the efficacy of RGA: it achieves 5.21%, 1.84% and 1.78% average improvements in 10 pre-training benchmarks across different model families. Moreover, we verify that pre-training with RGA leads to better post-training results, establishing a co-improving fly wheel between base and RL models.

## 2 BACKGROUND

**Standard LLM Pre-training.** LLMs are typically pre-trained using a next-token prediction objective (NTP). Consider a sequence of tokens, $x_{1:T} = [x_1, \ldots, x_T]$ over a vocabulary $\mathcal{V}$. An autoregressive model defines the probability of this sequence as the product of conditional probabilities:

$$p_\theta(x_{1:T}) = \prod_{t=1}^{T} p_\theta(x_t \mid x_{<t}). \tag{1}$$

Given a dataset $\mathcal{D}$ comprising $N$ sequences $\mathcal{D} = \{x_{1:T_i}^{(i)}\}_{i=1}^N$, the model parameters $\theta$ are optimized by minimizing the average negative log-likelihood, which assigns uniform weight to every token:

$$\mathcal{L}_{\mathrm{NTP}}(\theta) = -\frac{1}{\sum_i T_i} \sum_{i=1}^{N} \sum_{t=1}^{T_i} \log p_\theta \Big( x_t^{(i)} \mid x_{<t}^{(i)} \Big). \tag{2}$$

**Analyzing Log-Probability Discrepancies Between Base and RL Models.** We visualize the token-level log-probability gap between the base and RL models in Figure 2. For each model, we compute its probability distribution over the sentence drawn from the annealing corpus. The gap narrows substantially after annealing, indicating that the annealing stage is a critical turning point that drives a qualitative transformation of the base model.

To pinpoint where the RL model outperforms the pre-annealing base model on a specific sentence, Figure 3 visualizes the per-token prediction difference between the two models. For each token, the background color encodes the RL-to-base log-probability margin ($\Delta \log p$): lighter indicates a smaller difference, whereas deeper red indicates that the RL model assigns higher likelihood to the ground-truth token. Most tokens exhibit modest margins, while a few exhibit strong positive margins (deep red). This pattern suggests that the superior reasoning of the RL model is driven by a small set of pivotal tokens, with the rest largely following the distribution of the base model. Moreover, many high-margin tokens are discourse connectives (e.g., 'Therefore', 'So', 'gives'), format tokens (e.g. 'boxed'), or key verbs (e.g. 'follow', 'Adding'), consistent with the enhanced reasoning capabilities of RL models. For additional examples, please refer to Appendix B.

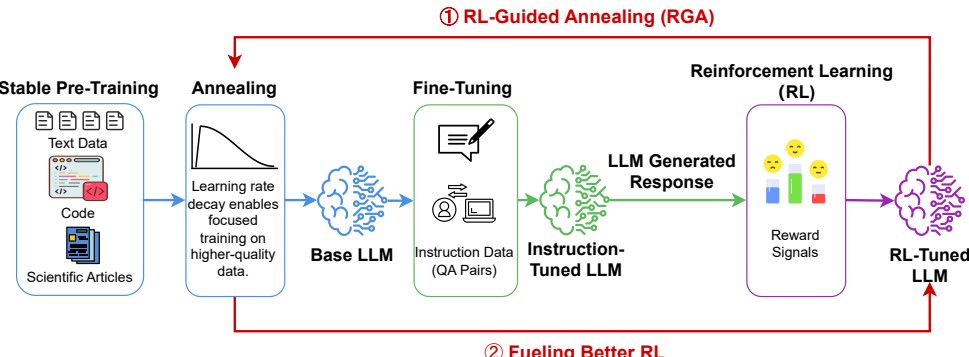

Figure 4: Overview of the proposed RL-Guided Annealing (RGA) pipeline. The procedure connects pre-training and post-training, yielding a positive feedback loop in which improvements from RL strengthen the base model, which in turn enhances subsequent RL.

Based on this observation, we attempt to assign larger weight to these important pivotal tokens during pre-training. The annealing phase is particularly well-suited for adjusting the token weighting within pre-training. This is because the data quality during annealing is significantly better, encompassing more instruction data, reasoning-related data, and similar content that is inherently more aligned with post-training objectives. Moreover, the annealing phase is the critical stage in which the model rapidly acquires higher-order capabilities (instruction following or reasoning) due to the rapid drop of learning rate. Therefore, adjusting token weights during this phase to emphasize pivotal tokens is a more efficient and effective strategy.

## 3 METHODOLOGIES

### 3.1 TOKEN REWEIGHTING MECHANISM

RGA dynamically adjusts token-level loss weights, enabling the base model to prioritize tokens fundamental to the reasoning improvements of RL model. Inspired by prior works (Lin et al., 2024; Zhu et al., 2025), RGA focuses on tokens that the current model has not yet effectively learned but have been mastered by the more sophisticated RL model. These tokens are characterized by a significant loss discrepancy between the two models. We formally define this discrepancy, termed the delta loss of token $x_t$, in equation 3.

$$L_\Delta(x_t) = L_{\text{Current}}(x_t) - L_{\text{RL}}(x_t) = -\log p_\theta(x_t \mid x_{<t}) + \log p_{\text{RL}}(x_t \mid x_{<t}). \quad (3)$$

Then we apply a sequence-level normalization to remove scale variation, as in equation 4, where $T$ denotes the sequence length:

$$\mu_\Delta = \frac{1}{T}\sum_{t=1}^{T} L_\Delta(x_t), \quad \widehat{L}_\Delta(x_t) = L_\Delta(x_t) - \mu_\Delta. \quad (4)$$

Intuitively, tokens with larger delta loss should receive higher weight. We therefore map the normalized deltas to non-negative, bounded weights via a clipped sigmoid in equation 5.

$$w_t = \text{clip}\Big(2 \cdot \sigma\big(\widehat{L}_\Delta(x_t)\big), 1 - \epsilon, 1 + \epsilon\Big), \quad \sigma(z) = \frac{1}{1 + e^{-z}}. \quad (5)$$

This clipping mechanism essentially serves as a regularization term, ensuring the loss does not deviate too far from the original log-likelihood maximization of the NTP paradigm and preventing over- or under-emphasis on any single token. $\epsilon$ is a hyperparameter with a default value of 0.2. We will further analyze the clipping mechanism in Section 4.3.

Finally, we compute a weighted loss by applying these weights to the standard next-token prediction objective, as defined in equation 2 and equation 6. By reweighting rather than discarding tokens, this

---

**Algorithm 1** RL-Guided Annealing (RGA)

---

**Require:** Current model $\theta$, frozen RL reference, clipping parameter $\epsilon$

1: **for** mini-batch sequences $x_{1:T}$ from the annealing corpus **do**

2: $\quad L_{\text{Current}}(x_t) \leftarrow -\log p_\theta(x_t \mid x_{<t}) \quad \forall t$ $\qquad\qquad\qquad$ ▷ NTP (equation 2)

3: $\quad L_{\text{RL}}(x_t) \leftarrow -\log p_{\text{RL}}(x_t \mid x_{<t}) \quad \forall t$ $\qquad\qquad\qquad\qquad$ ▷ added

4: $\quad w_t \leftarrow \text{clip}\Big(2 \cdot \sigma\big((L_{\text{Current}}(x_t) - L_{\text{RL}}(x_t)) - \mu_\Delta\big), 1 - \epsilon, 1 + \epsilon\Big)$ $\qquad$ ▷ added

5: $\quad \mathcal{L} \leftarrow \sum_t w_t \, L_{\text{Current}}(x_t)$ $\qquad\qquad\qquad\qquad\qquad\qquad$ ▷ RGA (equation 6)

6: $\quad$ update $\theta \leftarrow \theta - \eta \nabla_\theta \mathcal{L}$

7: **end for**

---

scheme preserves the semantic coherence of the sequence while directing the model's focus toward the most pivotal tokens during the annealing process.

$$\mathcal{L}_{\text{RGA}}(\theta) = -\frac{1}{\sum_i T_i} \sum_{i=1}^{N} \sum_{t=1}^{T_i} w_t^{(i)} \log p_\theta\left(x_t^{(i)} \mid x_{<t}^{(i)}\right). \tag{6}$$

The overall RGA algorithm is shown in Algorithm 1, which is very simple to incorporate with the current pre-training process.

### 3.2 RL-GUIDED ANNEALING

As shown in Figure 4, RGA is a simple yet effective paradigm that creates a self-reinforcing loop. It leverages the readily available last-round RL model from the training pipeline as the reference for annealing, thereby avoiding any extra training on manually curated clean datasets, whose construction is costly and often poorly defined. Moreover, the gains from RL transfer to the annealed base model, which in turn facilitates subsequent RL, yielding a self-reinforcing feedback loop: better base model will produce better RL model, and better RL model acts as better RGA reference.

On the other hand, we focus on the annealing stage for mainly two reasons. First, as the annealing corpus is largely reasoning oriented, adopting the latest RL model as the reference is well aligned with the data distribution. Second, compared with the stable pre-training stage with a stable or slowly decaying learning rate, the annealing stage delivers faster and more data-efficient gains, allowing RGA to translate the enhanced reasoning capabilities of RL into overall performance improvements more effectively.

## 4 EXPERIMENTS

### 4.1 EXPERIMENTAL SETUP

**Base Models.** We evaluate RGA across multiple representative model families that provide publicly available pre-annealed checkpoints, including OLMo-1B (OLMo et al., 2024) and SmolLM3-3B (SmolLM et al., 2025). Here, the *base model* refers to the pre-annealed version from official releases. Its corresponding publicly released RL-tuned version serves as the reference during annealing. To ensure a comprehensive evaluation, we additionally train a proprietary 2B experimental model (PropLM-2B) from scratch, as most open-source models, such as Qwen (Yang et al., 2025a) and Llama (Grattafiori et al., 2024), only provide post-annealed versions, with pre-annealed base models being relatively scarce. PropLM-2B follows a standard multi-head attention (MHA) architecture, with 32 layers, a hidden dimension of 2048, and 32 attention heads. It is stably pre-trained on 8 trillion tokens of general text data before undergoing RGA-based annealing experiments.

**Annealing Corpus.** For OLMo-1B, we employ the official OLMo annealing corpus (dolmino-mix-1124[1]), which consists of DCLM, FLAN, StackExchange Q&A, peS2o, Wikipedia/Wikibooks, and Dolmino Math. For SmolLM3-3B, we utilize their mixed dataset, which includes a combination of web text, code, and mathematical data. For PropLM-2B, we use a curated in-house multilingual

---

[1] https://huggingface.co/datasets/allenai/dolmino-mix-1124

Table 1: Few-shot accuracy across 10 widely used downstream tasks. Extended results and the num_shots used per task are provided in Appendix C. The best scores of each model family are **boldfaced**.

| Method | GSM8K | MATH | GPQA | BBH | IFE | HE | MBPP | TQA | ARC-C | MMLU$_{PRO}$ | Avg. |
|---|---|---|---|---|---|---|---|---|---|---|---|
| *Annealing on OLMo-1B* | | | | | | | | | | | |
| Pre-Annealed | 3.03 | 2.94 | 20.31 | 28.43 | 22.66 | 6.71 | 4.80 | 21.30 | 44.71 | 9.54 | 16.44 |
| Standard | 48.14 | 10.26 | 22.54 | 30.87 | 16.19 | 8.54 | 4.60 | 22.40 | 46.67 | 13.31 | 22.35 |
| RHO-1 | 50.42 | 10.32 | **25.45** | 29.33 | 19.06 | 6.71 | 6.20 | 23.38 | 46.42 | 13.68 | 23.10 |
| SGA | 50.34 | 12.16 | 23.44 | 31.15 | 13.07 | 10.98 | 9.00 | 24.11 | 47.70 | 15.23 | 23.72 |
| RGA | **61.64** | **14.50** | 24.55 | **32.07** | **28.54** | **12.80** | **9.20** | **25.58** | **49.23** | **17.44** | **27.56** |
| *Annealing on SmolLM3-3B* | | | | | | | | | | | |
| Pre-Annealed | 31.61 | 14.52 | **27.68** | 43.13 | 22.06 | 25.61 | 37.40 | 30.23 | **56.31** | 23.24 | 31.18 |
| Standard | **64.22** | 31.64 | 26.34 | 56.5 | 43.29 | 28.05 | 47.60 | 29.74 | 53.24 | 30.68 | 41.13 |
| RHO-1 | 62.93 | 28.70 | 26.56 | 55.32 | 38.73 | 31.10 | 43.80 | 28.52 | 54.10 | 28.19 | 39.80 |
| SGA | 62.77 | 31.24 | 24.11 | **58.65** | 45.08 | 36.59 | 49.20 | 30.97 | 53.84 | **30.83** | 42.33 |
| RGA | 63.76 | **31.68** | 26.12 | 58.27 | **45.68** | **37.20** | **49.60** | **31.95** | 54.69 | 30.73 | **42.97** |
| *Annealing on PropLM-2B* | | | | | | | | | | | |
| Pre-Annealed | 34.87 | 15.30 | 22.99 | 43.39 | 23.26 | 25.00 | 38.80 | **27.54** | 52.30 | 19.96 | 30.34 |
| Standard | 49.51 | **25.00** | 27.90 | 44.37 | 32.61 | 37.20 | 46.60 | 26.56 | 53.33 | 24.93 | 36.80 |
| RHO-1 | 42.46 | 16.44 | 26.79 | 40.45 | 32.01 | 27.44 | 39.20 | 26.68 | 49.23 | 21.69 | 32.24 |
| SGA | 47.54 | 19.84 | **30.13** | 44.60 | 32.73 | 32.93 | 43.40 | 25.46 | 52.65 | 24.42 | 35.37 |
| RGA | **52.69** | 24.50 | 29.69 | **47.21** | **36.93** | **39.94** | **47.00** | 27.42 | **54.61** | **25.85** | **38.58** |

annealing corpus comprising web text, encyclopedic material, mathematical and STEM reasoning, as well as code. The annealing corpus for all three models contains 50B tokens. The sequence length is 4K for OLMo-1B and 8K for both SmolLM3-3B and PropLM-2B.

**Annealing Setting.** For OLMo-1B, SmolLM3-3B and PropLM-2B, we employ maximum learning rates of 3e-4, 2e-4 and 2e-4, respectively. These rates follow a cosine decay schedule to a minimum learning rate of 1e-7. The global batch sizes are set to 2M, 2M and 64M tokens, respectively, matching those used during each model's stable pre-training phase. For the clipping parameter in equation 5, we set $\epsilon = 0.2$.

**Baselines.** We compare RGA with four baselines:

- **Pre-Annealed.** The pre-annealed base model, which is the starting point of the annealing phase.

- **Standard.** The conventional next-token prediction objective that assigns uniform weights to all tokens, as shown in equation 2.

- **RHO-1 (Lin et al., 2024).** A token-level data-selection baseline that retains tokens with high RHO loss during training and discards the remainder. Following their methodology, we carefully construct a 50B high-quality dataset from our proprietary data. We then continue pre-training the same base model on this dataset to derive the reference model used in our experiments. The top-$k$ ratio is set to 0.8.

- **SGA (SFT-Guided Annealing).** An ablation study to evaluate the impact of replacing the RL reference model with a SFT model for guiding the annealing process.

**Evaluation.** To comprehensively evaluate the pre-trained models, we assess their few-shot performance across multiple widely-used downstream tasks using the lm-eval-harness framework (Gao et al., 2021). To further demonstrate the superiority of RGA, we also conduct subsequent post-training—SFT, DPO, and RLVR[2]—and evaluate the zero-shot accuracy of the resulting models. Further evaluation details can be found in Section C, with extended results provided in Appendix D.

---

[2]**SFT**: supervised fine-tuning; **DPO**: direct preference optimization; **RLVR**: reinforcement learning with verifiable rewards.

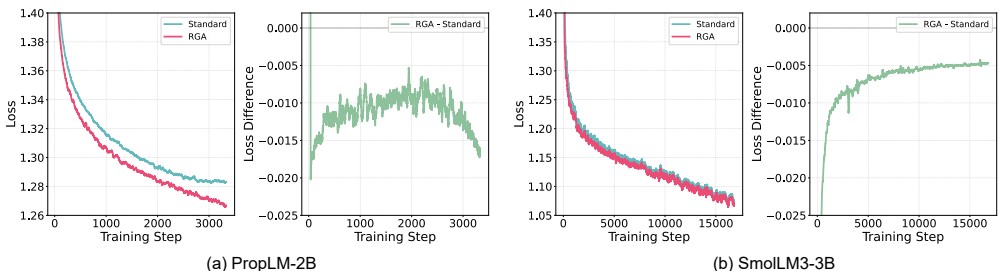

(a) PropLM-2B        (b) SmolLM3-3B

Figure 5: Language modeling loss over the annealing process. We also plot the loss difference as RGA loss minus Standard loss to aid visualization and interpretability.

## 4.2 MAIN RESULTS

**RGA Enhances Pre-Training Performance.** Table 1 reports few-shot accuracies on downstream tasks for LMs trained with different methods, from which we have the following observations. *First*, annealing proves to be a critical stage: across the three model families, all annealed variants significantly outperform the pre-annealed base. *Second*, RGA achieves the best performance on most tasks by dynamically adjusting token-level weights to focus on those most pivotal to the improved reasoning capabilities of the RL model. *Finally*, RGA surpasses RHO-1 by directly using the RL counterpart as reference, avoiding costly data curation and teacher training with uncertain returns. Figure 5 shows the loss trajectories during the annealing process for PropLM-2B and SmolLM3-3B. The loss of RGA is consistently lower than that of the standard approach.

**Gains in Pre-Training Transfer to Post-Training.** RGA enhances performance not only during the annealing stage but also provides a superior foundation for subsequent post-training. To demonstrate the advantage of our approach, we further conduct post-training experiments. As illustrated in Figure 6, we apply identical post-training procedures to both the standard base model and the RGA base model on OLMo-1B. We observe that *the performance gains achieved during pre-training transfer consistently to post-training*, regardless of the specific technique employed. Figure 6 (a) presents the results of applying RL directly to the annealed base model, while figure 6 (b) shows the results after applying the full post-training pipeline, including SFT, DPO, and RL. Additional results are provided in the Appendix D.

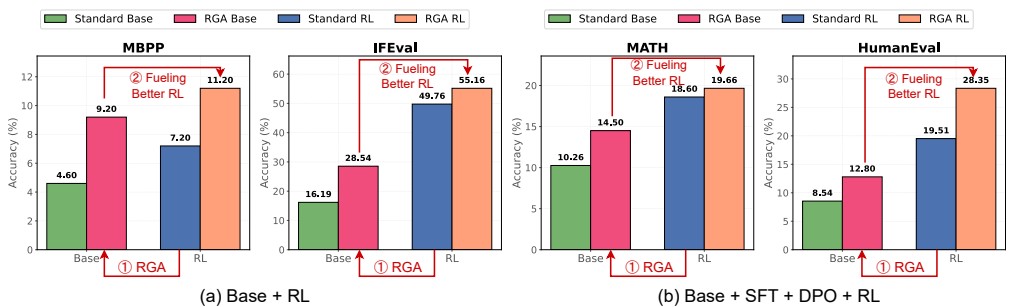

(a) Base + RL               (b) Base + SFT + DPO + RL

Figure 6: Performance gains of RGA on OLMo-1B during pre-training transfer consistently to post-training, regardless of the post-training process (SFT, DPO, or RL). Figure (a): applying RL directly to the annealed base model. Figure (b): applying the complete post-training procedure to the annealed base model.

**Co-Improving Feedback Loop Between Pre-Training and Post-Training.** We have demonstrated that RGA enhances performance during both pre-training and post-training. We now show that *a stronger RGA RL model can improve the annealing process even further*, as illustrated in Figure 7. Here, the $RGA^2$ base model is produced by reusing the RGA RL model as a new reference during annealing. This creates a cascading effect of improvements: RL-guided annealing produces a

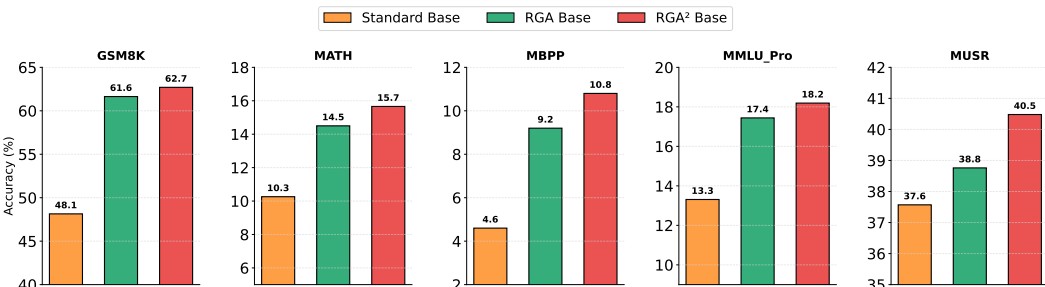

Figure 7: Co-improving feedback loop between pre-training and post-training via RGA. *RGA base* is the base model after RL-guided annealing; *RGA RL* is the RL-tuned model trained on RGA base; $RGA^2$ *base* reuses the RGA RL as the new reference for the annealing stage. Improvements cascade: RL-guided annealing → stronger RGA base → stronger RGA RL → stronger $RGA^2$ base.

stronger RGA base, which in turn yields a stronger RGA RL model, leading to an even stronger $RGA^2$ base. As shown, the $RGA^2$ base outperforms the RGA base, demonstrating that RGA is a robust technique capable of enhancing the entire LLM training pipeline.

## 4.3 ANALYSIS AND DISCUSSION

**Choosing the Reference: Why RL over SFT.** In RGA, we employ the RL model as the reference model. To further investigate the influence of reference model selection, we conduct an ablation study using a SFT model as an alternative reference. Figure 8(a) visualizes token-level preferences of the SFT and RL models. The background is color-coded by the log-probability difference for each ground-truth token, $\Delta \log p = \log p_{\text{RL}} - \log p_{\text{SFT}}$. Deeper red indicates tokens where the RL model assigns higher likelihood, while a deeper blue indicates tokens favored by the SFT model. This color mapping provides an intuitive way to identify where and how the prediction patterns of these two models diverge. We observe that the RL reference assigns higher likelihood to discourse connectives (e.g., 'Therefore', 'then', 'since', 'so'), consistent with its enhanced reasoning capabilities, while SFT tends to favor content words such as 'cube'. More examples are provided in Appendix B.

Figure 8(b) plots downstream performance during annealing on OLMo-1B when guided by SFT (SGA) versus RL (RGA), where RGA consistently outperforms SGA over training. Table 1 shows the same trend across additional benchmarks, supporting the choice of the RL model as the reference.

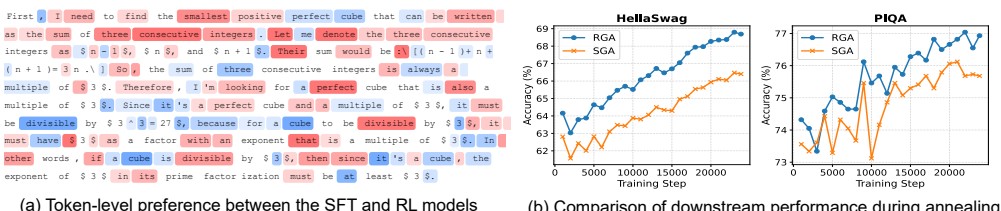

(a) Token-level preference between the SFT and RL models

(b) Comparison of downstream performance during annealing

Figure 8: Comparison between SGA and RGA. (a) Background color encodes the log-probability margin $\Delta \log p = \log p_{\text{RL}} - \log p_{\text{SFT}}$ for the ground-truth token: deeper red denotes RL-favored tokens (higher likelihood), deeper blue denotes SFT-favored tokens, and white indicates negligible differences ($|\Delta \log p| < 0.1$). (b) Downstream performance during annealing on OLMo-1B.

**Impact of the Clipping Mechanism.** We now examine the role of the clipping mechanism introduced in equation 5. As illustrated in Figure 9, applying the clipping mechanism to RGA elevates the overall gradient norm. This increase indicates that clipping prevents certain token weights from becoming diminutive, thereby mitigating token under-emphasis. As a result, the model can learn more effectively, achieving a more rapid loss reduction. For comparison, we also present the loss and gradient norm trajectories for the RHO-1 baseline. These results show that RHO-1's strategy

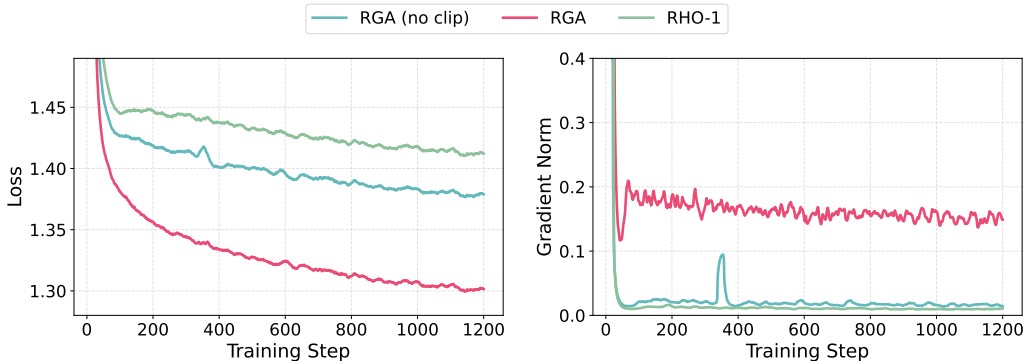

Figure 9: Clipping mechanism in RGA. By bounding token-level weights, clipping keeps gradient norms within a normal range and accelerates loss reduction.

of directly discarding tokens not only impairs semantic coherence but also yields extremely low gradient norms due to token under-emphasis, which hinders efficient loss reduction.

## 5 RELATED WORK

**Data Selection Strategies in Pre-Training.** The quality and composition of the training corpus are critical factors in pre-training LLMs. A primary objective is to optimize this corpus through careful data selection, which is crucial for developing the model's foundational capabilities (Xie et al., 2023b; Albalak et al., 2024; Liu et al., 2024; Gu et al., 2024; Zhu et al., 2025). Common strategies employ lightweight filters, including heuristic-based, classifier-based (Mann et al., 2020), and perplexity or loss-based approaches (Qin et al., 2023; Wenzek et al., 2019). More advanced techniques utilize reference models as proxies for data selection. Ankner et al. (2024) select examples with reference model sequence-level log-perplexities within specified ranges. Mindermann et al. (2022) select examples and Lin et al. (2024) select tokens for training based on excess training loss over a reference model. However, our proposed RGA offers a token-level reweighting approach rather than hard selection, and directly leverages the existing RL model in the training pipeline as a reference, eliminating the need for additional training costs of the reference model.

**Token-Level Analysis.** Recently, numerous studies have performed token-level analysis in post-training, especially in reinforcement learning with verifiable rewards (RLVR), focusing on token entropy patterns. Yang et al. (2025b) argue that low-probability tokens disproportionately influence model updates due to their large gradient magnitudes, which hinders effective learning of LMs. Wang et al. (2025a) observe that RLVR largely adheres to the base model's entropy patterns, primarily adjusting the entropy of a small fraction of high-entropy tokens, which act as critical forks that steer the model toward diverse reasoning pathways. Arbuzov et al. (2025) share a similar perspective, noting that LLM errors are not uniformly distributed but are concentrated at sparse "key tokens" representing critical decision junctions. Wang et al. (2025b) propose an adaptive weighting strategy designed to prioritize uncertain data during post-training.

## 6 CONCLUSION

This work investigates methods for coupling pre-training and post-training, identifying the annealing (mid-training) phase as a critical turning point characterized by substantial distribution shift. Building on this insight, we introduce RL-Guided Annealing (RGA), a method that leverages the RL counterpart as the reference model during annealing to assign dynamic, token-wise weights. These weights guide the base model toward tokens that are most pivotal for the significant reasoning improvements demonstrated by RL models. RGA reweights tokens instead of discarding them, thereby preserving semantic coherence while largely maintaining the efficiency of standard next-token prediction. Evaluations across multiple model families show that RGA consistently improves performance on pre-training benchmarks and further enhances post-training outcomes, establishing a co-improving fly wheel between base and RL models.

REPRODUCIBILITY STATEMENT

To facilitate the full reproducibility of our work, we provide comprehensive details of our methodology and experimental configuration. The pseudocode detailing our proposed RGA is included in Algorithm 1. Section 4.1 and Appendix C contain all relevant implementation details, such as data construction procedures, model specifications, hyperparameter settings, and evaluation details. Our code will be made publicly available to support replication and further research.

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

# A  THE USE OF LLMS

Use of large language models (LLMs) in this study was limited exclusively to language refinement. The models were used to improve clarity, correct grammar, and assist with translation to enhance the manuscript's readability and accessibility. They were not used for research ideation, data analysis, or the generation of any core scientific content.

# B  VISUALIZATION OF LOG-PROBABILITY DISCREPANCIES BETWEEN DIFFERENT MODELS

## B.1  PRE-ANNEALED BASE MODEL VERSUS RL MODEL

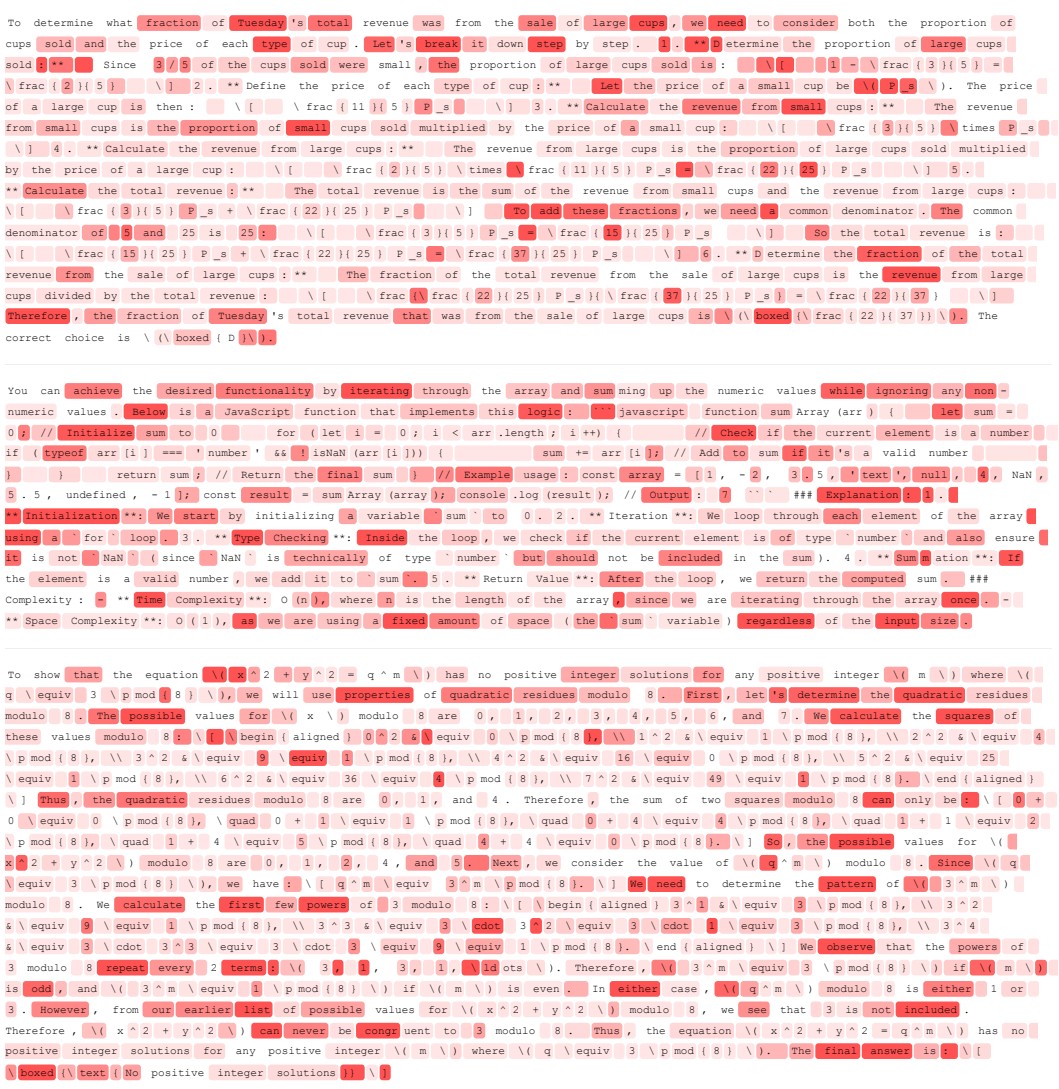

Figure 10: Token distribution divergence between the base and RL models. Background intensity encodes the per-token log-probability margin ($\Delta \log p = \log p_{\text{RL}} - \log p_{\text{base}}$), with deeper red indicating higher confidence in the ground-truth token from the RL model.

To identify where the RL model surpasses its pre-annealed base model on a given sentence, Section 2 presents an example visualizing log-probability discrepancies between the base and RL models.

Figure 10 provides additional examples for further analysis. For each token, the background color encodes the RL-to-base log-probability margin ($\Delta \log p$): lighter shades indicate smaller margins, whereas deeper red indicates that the RL model assigns higher likelihood to the ground-truth token. Most tokens show modest margins, while a small subset exhibits large positive margins (deep red). This pattern suggests that the RL model's superior reasoning is driven by a limited set of pivotal tokens, with the remainder largely following the base model's distribution.

## B.2 SFT Model versus RL Model

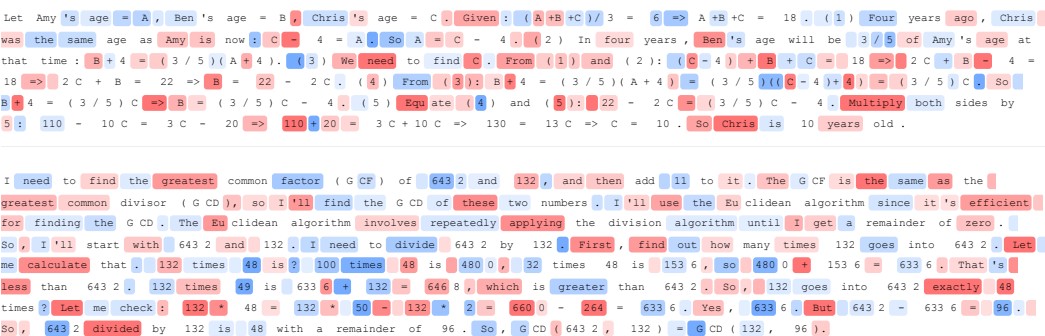

Figure 11: A comparison of token-level preferences between the RL and SFT models. The background color of each ground-truth token indicates the log-probability margin $\Delta \log p = \log p_{\text{RL}} - \log p_{\text{SFT}}$. Deeper red signifies a stronger preference from the RL model, deeper blue from the SFT model, and white denotes a negligible difference ($|\Delta \log p| < 0.1$).

In this section, we analyze the rationale for selecting the RL model over the SFT model as a reference during the annealing process. A key justification lies in their distinct token-level preferences, which we now examine in greater detail. As our examples illustrate, the RL model assigns a significantly higher likelihood to discourse connectives (e.g., 'First', 'But', 'So', 'Let'). This preference, visualized by a deeper red background for the corresponding tokens, is consistent with the enhanced reasoning capabilities cultivated during reinforcement learning, making the RL model a more suitable reference.

Table 2: Zero-shot accuracy of post-trained models on OLMo-1B across 11 widely used downstream tasks. The best scores are **boldfaced**.

| Method | GSM8K | MATH | MATH-500 | MBPP | MBPP+ | HE | HE+ | ARC-C | GPQA_DM | MMLU_PRO | IFE | Avg. |
|---|---|---|---|---|---|---|---|---|---|---|---|---|
| Stage: SFT | | | | | | | | | | | | |
| Standard | 42.61 | 12.28 | **11.40** | 16.20 | 25.66 | 21.95 | 18.90 | 40.96 | 24.75 | **14.34** | 56.83 | 25.99 |
| RGA | **47.00** | **16.54** | 10.40 | **19.60** | **29.10** | **26.22** | **21.95** | **41.38** | **25.76** | 13.56 | **59.11** | **28.24** |
| Stage: DPO | | | | | | | | | | | | |
| Standard | **57.70** | 15.10 | **16.00** | 9.20 | 18.78 | 24.39 | 21.95 | 43.43 | 21.72 | 14.61 | 67.39 | 28.21 |
| RGA | 55.04 | **17.96** | 15.00 | **12.40** | **21.69** | **25.61** | **22.56** | **45.22** | **23.23** | **15.78** | **69.18** | **29.42** |
| Stage: RL | | | | | | | | | | | | |
| Standard | **66.72** | 18.60 | **21.40** | 8.80 | 17.33 | 19.51 | 17.07 | 43.00 | 20.20 | 14.58 | 69.90 | 28.83 |
| RGA | 65.73 | **19.66** | 20.20 | **11.60** | **22.09** | **28.35** | **24.39** | **44.80** | **25.25** | **16.73** | **70.62** | **31.77** |

## C  Evaluation Details

### C.1  Evaluation of Pre-trained Base LMs

We comprehensively evaluate the pre-trained base models by measuring their few-shot performance across a suite of widely used downstream benchmarks, including: GSM8K (Cobbe et al. (2021);

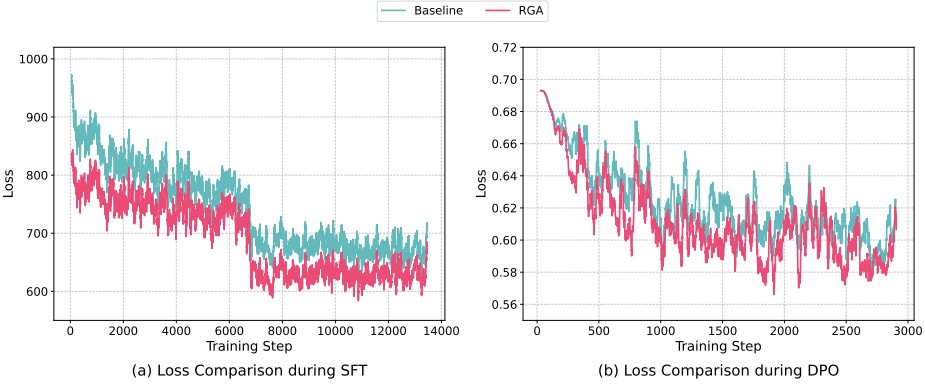

Figure 12: Loss trajectories during SFT and DPO on OLMo-1B.

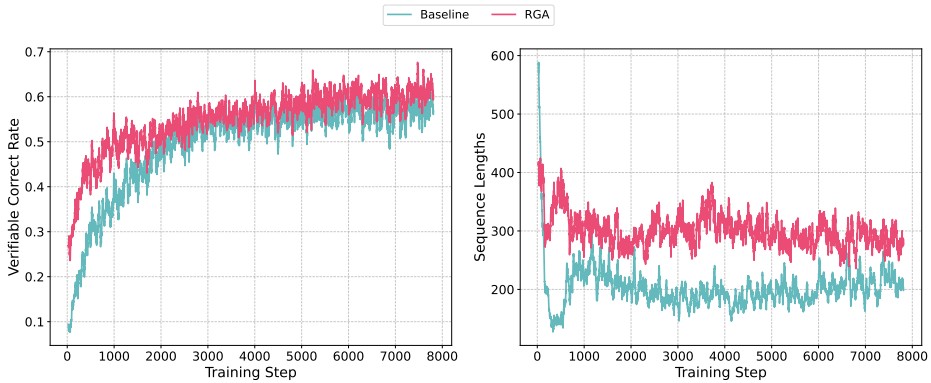

Figure 13: Performance comparison at the reinforcement learning stage on OLMo-1B.

8-shot), Minerva Math (Lewkowycz et al. (2022); 4-shot), GPQA (Rein et al. (2024); 5-shot), Big-BenchHard (BBH; Suzgun et al. (2022); 3-shot CoT), IFEval (IFE; Zhou et al. (2023) 3-shot), HumanEval (HE; Zheng et al. (2023); 0-shot), MBPP (Austin et al. (2021); 3-shot), TruthfulQA (TQA; Lin et al. (2021); 3-shot), ARC-Challenge (Clark et al. (2018); 25-shot), and $MMLU_{Pro}$ (Wang et al. (2024); 5-shot). We report Pass@1 for MBPP and HumanEval. In extended experiments, we also evaluate additional benchmarks such as HellaSwag (Zellers et al., 2019), PIQA (Bisk et al., 2020) and MuSR (Sprague et al., 2023).

## C.2 EVALUATION OF POST-TRAINED LMS

Given that the post-trained models already exhibit instruction-following ability, we evaluate their zero-shot accuracy. We report results on 11 widely used benchmarks: GSM8K (Cobbe et al., 2021)), Minerva Math (Lewkowycz et al., 2022), Math-500 (Lightman et al., 2023), MBPP (Austin et al., 2021), MBPP+ (Liu et al., 2023), HumanEval (HE; Zheng et al. (2023)), HumanEval+ (HE+; Liu et al. (2023)), ARC-Challenge (Clark et al., 2018), $GPQA_{Diamond}$ ($GPQA_{DM}$; Rein et al. (2024)), $MMLU_{Pro}$ (Wang et al., 2024), IFEval (IFE; Zhou et al. (2023)).

## D MORE POST-TRAINING RESULTS

To further demonstrate that RGA enhances performance not only during annealing but also in subsequent stages, we perform additional post-training on the annealed base models. As illustrated in Figure 12, the loss trajectories for both SFT and DPO show that RGA consistently maintains lower

loss values compared to the standard approach. Furthermore, we present a performance comparison of the reinforcement learning process in Figure 13, which indicates that RGA achieves a higher verifiable correct rate while also maintaining longer sequence lengths.

In Table 2, we present the performance of the post-trained models across 11 downstream benchmarks. The results demonstrate that the performance gains achieved during the annealing stage are effectively maintained throughout the entire post-training process.

