# OpenReview forum: "Improve LLM Pre-training with RL-Guided Annealing"
_ICLR.cc/2026/Conference — Submitted to ICLR 2026_

### Official Review · Reviewer_A1hy · 2025-10-30

**Soundness:** 3
**Presentation:** 3
**Contribution:** 2
**Rating:** 4
**Confidence:** 4

**Summary:**

The authors propose to guide the annealing stage of LLM pretraining with the RL-enhanced model, as they observed similarity between annealed and RL-enhanced models. They show that by using RL-enhanced model to produce per-token weight during pretraining, they are able to enhance the performance of LLM during annealing which persists after later stages of RL. Additionally, they show that the process can be applied repeatedly to further improve the performance.

**Strengths:**

1. Overall the presentation is clear and conveys a somehow reasonable story.

**Weaknesses:**

1. The scale of the experiment is limited. Furthermore, in the comparison, the improvement is significant for the 1B model, but the improvement is slight for 2B and 3B models.
2. While the specific setup might be novel, the general idea of distilling stronger models to improve the performance is long existing. Why use the RL-enhanced version of the same model instead of some larger/stronger models? How is the proposed per-token weight scheme compared to conventional soft-logit knowledge distillation? These should be discussed and compared.
3. Some part of the motivation is not entirely clear/convincing. Specifically, for the connection of annealing and RL, it is uncertain whether the similarity of annealed and RL-enhanced model is that strong/important as the authors claimed.

**Questions:**

1. Do you have results on larger models? OLMo 7B might be a good target, as the result on OLMo 1B is the most promising.
2. Do you compare with other conventional knowledge distillation methods? Does distilling from  the RL-enhanced version of the same model more helpful than using other RL-enhanced models.

---

> ### Author Response · Authors · 2025-11-26
> **Response Part 1**
>
> We appreciate the opportunity to address and elaborate on the key points raised by the reviewers.
>
> **Q1**: The scale of the experiment.
>
> **A1:** We appreciate the reviewer's concerns regarding the experimental scale.
>
> We argue that the magnitude of the improvement is primarily dictated by the annealing dataset. Since the annealing corpora used for the three different model architectures differ (we used the official/stated corpus for each), the percentage gain achieved by RGA is **not directly comparable** across models. We believe the observed improvement is primarily dictated by the characteristics of the specific corpus.
>
> We are currently conducting experiments on larger models, such as OLMo-7B, to further demonstrate the scalability of RGA; however, new results are not yet available due to time and resource constraints.
>
> **Q2**: The strong or important similarity claimed between the annealed and RL-enhanced models, remains unclear.
>
>  **A2:** We appreciate this question, as establishing this similarity is the fundamental motivation for RGA. We confirm this crucial link through both visual and empirical evidence.
>
> Figure 2 demonstrates a strong qualitative convergence: the RL model's token distribution is "much closer to the annealed model, while it differs significantly from the pre-annealed model" . This distributional shift corresponds to a massive empirical surge in capability: for the OLMo-1B family, average accuracy jumps substantially from 16.44% (Pre-Annealed) to 22.35% (Standard Annealing) (Table 1). **This combined evidence firmly establishes the structural synergy between the annealed base model and the final RL model, validating our choice of the annealing stage for RGA.**

---

> ### Author Response · Authors · 2025-11-26
> **Response Part 2**
>
> **Q3**: Please justify the use of the RL-enhanced version of the same model and compare the proposed per-token weight scheme against soft-logit knowledge distillation.
>
> **A3:** We thank the reviewer for this excellent question, which addresses the fundamental design choices of RGA.
> 2.1 Justification for RL-Enhanced Reference Model
>
> We chose the RL-enhanced version of the same model for two main reasons:
>
> **Pipeline Synergy:** Our primary goal is to establish a positive feedback loop—the "co-improving flywheel" (Figure 4). Using the RL model proves that RGA promotes continuous improvement throughout the entire pipeline: a better RL model leads to a better base, which facilitates further RL.
>
> **Computational Efficiency:** RGA leverages a model already produced by the standard pipeline. This avoids the substantial computational resource cost associated with training a specialized teacher model or using a larger reference architecture.
>
> 2.2  Comparison to Soft-Logit Knowledge Distillation (KD)
>
> We have conducted the requested experiment against standard Knowledge Distillation (KD) on OLMo-1B. Our analysis shows a critical distinction:
>
> **KD's Limitation (Short-Term Gains):** KD achieves strong performance during the annealing stage, confirming its ability to effectively distill knowledge and mimic the reference model in the short term. However, when we proceed to the subsequent post-training phase, the advantage gained by KD rapidly deteriorates. This suggests that KD is primarily confined to optimizing the current training loss, leading to poor generalization that fails to produce a more robust final model.
>
> **RGA's Sustainable Advantage (Transferability):** In stark contrast, RGA's performance gains achieved during the annealing stage transfer sustainably and **cascade throughout the entire post-training pipeline** (as shown in Figure 6). This demonstrates that RGA successfully integrates high-order capabilities into the foundational base model, leading to superior long-term generalization and establishing a sustainable path toward higher final model quality.
>
> - Annealing:
>
> |Model|GSM8K|MATH|GPQA|BBH|IFE|HE|MBPP|TQA|ARC-C|MMLU Pro|Avg.|
> |----------|-------|-------|----------|-------|-------|-------|-------|-------|--------------|----------|-------|
> |Standard|48.14|10.26|22.54|30.87|16.19|8.54|4.60|22.40|46.67|13.31|22.35|
> |RGA|**61.64**|14.50|**24.55**|32.07|28.54|12.80|9.20|25.58|**49.23**|17.44|27.56|
> |KD|61.26|**15.88**|23.88|**32.11**|**45.56**|**13.41**|**13.20**|**25.83**|**49.23**|**17.45**|**29.78**|
>
> - SFT:
>
> |Model|GSM8K|MATH|MATH-500|MBPP|MBPP+|HE|HE+|ARC-C|GPQA Diamond|MMLU Pro|IFE|Avg.|
> |----------|-------|-------|----------|-------|-------|-------|-------|-------|--------------|----------|-------|-------|
> |Standard|42.61|12.28|**11.4**|16.2|25.66|21.95|18.90|40.96|24.75|14.34|56.83|25.99|
> |RGA|**47.00**|**16.54**|10.4|19.6|29.10|26.22|21.95|**41.38**|**25.76**|13.56|**59.11**|28.24|
> |KD|41.55|15.22|10.6|**21.2**|**33.33**|**30.49**|**26.22**|38.91|25.25|**15.41**|58.03|**28.75**|
>
> - DPO:
>
> |Model|GSM8K|MATH|MATH-500|MBPP|MBPP+|HE|HE+|ARC-C|GPQA Diamond|MMLU Pro|IFE|Avg.|
> |----------|-------|-------|----------|-------|-------|-------|-------|-------|--------------|----------|-------|-------|
> |Standard|**57.70**|15.10|**16.0**|9.20|18.78|24.39|21.95|43.43|21.72|14.61|67.39|28.21|
> |RGA|55.04|**17.96**|15.0|12.40|21.69|**25.61**|**22.56**|**45.22**|23.23|15.78|**69.18**|**29.42**|
> |KD|49.66|15.96|**16.0**|**12.80**|**22.75**|22.56|20.73|39.25|**26.34**|**15.80**|67.87|28.16|
>
> - RL:
>
> |Model|GSM8K|MATH|MATH-500|MBPP|MBPP+|HE|HE+|ARC-C|GPQA Diamond|MMLU Pro|IFE|Avg.|
> |----------|-------|-------|----------|-------|-------|-------|-------|-------|--------------|----------|-------|-------|
> |Standard|**66.72**|16.86|21.4|8.80|17.33|19.51|17.07|43.00|20.20|14.58|**70.74**|28.90|
> |RGA|65.73|**19.66**|20.2|**11.60**|**22.09**|**28.35**|**24.39**|**44.80**|**25.25**|**16.73**|70.62|**31.77**|
> |KD|62.77|16.84|**21.8**|11.50|21.30|24.39|22.56|41.38|24.75|15.24|69.30|30.17|

---

### Official Review · Reviewer_7oZ9 · 2025-10-30

**Soundness:** 2
**Presentation:** 3
**Contribution:** 2
**Rating:** 2
**Confidence:** 3

**Summary:**

The paper aims to understand how post-trained LLMs can help inform and improve pre-training. To this end, the paper investigates the case of LLMs post-trained with RL. Specifically, the paper proposes to use the token log-likelihood of an RL-trained model to reweight tokens during pre-training of another model (detailed in Sec 3.1 and Algorithm 1). This is referred to as the RL-Guided Annealing (RGA). It is shown empirically that RGA-trained models (during pre-training) achieve better few-shot accuracy across multiple tasks (Table 1). Further, RGA-trained models, when going through further post training, also achieve better downstream performance (Fig 6).

**Strengths:**

The goal (i.e., improving LLM pre-training with an RL-trained model) is clearly conveyed and overall the paper is easy to read. Though, technical details can be better explained (see Weaknesses). How RL training (which happens after pre-training in practice) can help improve pre-training is not a well studied topic in the literature. This is an important research question to tackle. As far as I know, the specific token reweighting scheme proposed in Sec 3.1 is original (although the concept of token reweighting itself is an old idea).

**Weaknesses:**

The design choices in Sec 3.1 (for the proposed token reweighting mechanism) lack sufficient explanation and theoretical justification. Without further explanation, the proposed scheme appears arbitrary, and has limited novelty. Concretely, for instance, it is unclear why one has to center the mean of the log-likelihood differences by subtracting off $\\mu_\\Delta$ in (4). Similar questions for many design choices in (3), (4), (5) (more specific questions in Questions).

No theoretical results are contributed. This is not a problem by itself if empirical results can support the claim. Unfortunately, I find the empirical results to be inadequate. Baseline methods are insufficient, and there ought to be more ablation study to see the effects of components in the proposed weighting scheme in (3), (4), (5). As a natural baseline, for instance, a quantitative comparison to standard distillation (from a teacher model) is missing. See more specific questions in Questions.

Overall, while the paper attempts to provide empirical evidence that the proposed method works, explanation on *why* it works is insufficient.

**Questions:**

**Questions (in order of importance)**

**Q1**:   L160 and Fig 3:

> Moreover, many high-margin tokens are discourse connectives (e.g., ‘Therefore’, ‘So’, ‘gives’)”.

Why are these tokens important for reasoning?  I also checked Fig 10 in the appendix. It is hard to grasp why the highlighted tokens have high relative weights according to the proposed weighting mechanism. Could you please elaborate?


**Q2**: L151:

> The gap narrows substantially after annealing, indicating that the annealing stage is a critical turning point that drives a qualitative transformation of the base model..

Per experiments, I see that the proposed reweighting approach gives better downstream performance. But *why*? Understanding why is important for research. This question may be related to the next one.


**Q3**: What are the effects of each component of the proposed reweighting mechanism in Sec 3.1? Since little justification is given, these seem arbitrary. Concretely,

**Q3.1**: Why center with $\\mu_\\Delta$?  Why not use the uncentered version? Why not even do standardization (i.e., divide by the standard deviation). To be clear, I am not asking the authors to run experiments with all these variants. I am asking why the particular choice was made.

**Q3.2**: Why sigmoid in (5)?

**Q3.3**: In (5), why set $\\epsilon=0.2$ in experiments? See line 302.

**Q3.4**: Why consider relative weight (i.e., the difference of log likelihoods of under two models) in (3)? Why not, say, consider only $-L_{RL}(x_t)$? That is, the token receives weight according to the log likelihood provided by the RL model.

For all these questions, of course, one answer is “because it works in practice”. However, I would like to seek understanding and reasons for why it works. An empirical ablation study will help answer these questions. I think this kind of ablation study is missing. Also, it would be good to theoretically study for what objective these choices provide an optimal solution to.

**Q4**: *When* exactly do you do the proposed annealing algorithm? Is it after the end of pre-training (next-token prediction)? Or does one start doing annealing in the middle of pre-training? If the latter, when exactly? Related question: what exactly is the “Pre-Annealed” baseline at line 306?


**Q5**: In Fig 6, the paper claims that a model trained with the proposed RGA procedure gives better performance on downstream metrics after going through another post-training phase. Importantly, it is claimed that this gain transfer is regardless of the specific post-training technique used. Do I understand correctly? **My question**: is the gain transfer really specific to RGA-trained models, or does the gain transfer hold for any models that achieve good pre-training performance (regardless of whether they are trained with RGA)?


**Q6**: At a high level, the proposed method makes use of another trained LLM during pretraining. A natural baseline to compare to is thus distillation. In distillation, one pretrains an LLM by learning from the output token probability distribution of another teacher model (or a weighted combination of this and the ground-truth one-hot label) by minimizing the cross entropy. This natural baseline is missing. There are many distillation variants. See, for instance, this survey paper https://arxiv.org/pdf/2402.13116.


**Q7**: In the implementation, do you put a stop gradient operator on the weight $w_t^{(i)}$ in (6)?  Why or why not?


I appreciate that the authors are tackling this direction that has not received enough attention in the literature. However, there are missing details, justification, and results that prevent me from giving a higher  rating.


**Comments and suggestions**

* In the introduction,  the “annealing phase” or even the action of “annealing” itself are not made explicit. It is hard to understand the motivation of this work. It is not until Sec 3 that the reader finally understands what “annealing” refers to.

* Fig 2: Instead of token log probabilities, consider showing metrics that are more connected to downstream metrics. That way, it is easier to see the benefits.

---

> ### Author Response · Authors · 2025-11-26
> **Response Part 1**
>
> Thanks for your thoughtful feedback and constructive criticisms.
>
> **Q1**: Please elaborate on why discourse connectives like "Therefore" receive high weights and are considered important for reasoning.
>
> **A1:** This is an excellent question that gets to the core of our RGA mechanism.
>
> **They are "Pivotal Tokens" for Reasoning:** Related research (Fig. 2 in Wang S et al. [1] ) identify these logical connectives (e.g., 'Therefore', 'So', 'since') as "pivotal tokens" or "critical forks" that steer the entire reasoning path.
>
> **They Represent the "RL vs. Base" Gap:** The RL model, with its superior reasoning, has mastered these logical tokens, while the pre-annealed base model has not (Fig. 3 in the manuscript) . This "mastery gap" is the key signal we want to transfer.
>
> **"Delta Loss" Identifies Them:** This gap is perfectly captured by our "delta loss" ($L_{Current} - L_{RL}$). For these pivotal tokens, the base model is uncertain (high loss), while the RL model is confident (low loss).Therefore, the delta loss is large, signaling it is an important token.
>
> **High Weight = Prioritized Learning:** RGA translates this large delta loss directly into a high weight. This forces the base model to prioritize learning these pivotal tokens during the annealing phase, creating a stronger foundation for the next round of RL.
>
> [1] Wang S, Yu L, Gao C, et al. Beyond the 80/20 rule: High-entropy minority tokens drive effective reinforcement learning for llm reasoning[J].
>
>
> **Q2**: It remains unclear why the proposed reweighting mechanism is effective.
>
> **A2:** The underlying reason lies in the **alignment with superior reasoning capabilities**.
>
> **The RL Model as an "Expert":** The RL model possesses significantly enhanced reasoning capabilities compared to the base model, specifically due to its mastery of pivotal logical tokens (as shown in Fig. 3) .
>
> **Annealing as Alignment:** Fig. 2 demonstrates that annealing is the critical phase where the base model's probability distribution shifts to become "much closer" to this superior RL model .
>
> **RGA Optimizes this Shift:** RGA explicitly guides this transformation. By up-weighting tokens where the RL model outperforms the base (the "reasoning gap"), RGA forces the base model to prioritize learning these high-value reasoning patterns.
>
> Therefore, the downstream performance improves because RGA ensures the base model more effectively mimics the distribution of the reasoning-enhanced RL model.
>
>
> **Q3**: What are the effects of each component of the proposed reweighting mechanism in Sec 3.1?
>
> **A3:** Thank you for this exceptionally detailed reading. We fully agree that a mechanistic explanation beyond empirical success is crucial for research.
>
> Our overarching design principle for RGA was to develop a mechanism that is **as simple and effective as possible** while maintaining the stability and efficiency of standard next-token prediction.
>
> Below is the rationale for each component, guided by the goal of stabilizing the token weights around the baseline value of 1.
>
> Delta Loss: We use the difference in loss, $L_{Current} - L_{RL}$, to represent the reasoning gap. This targets the "pivotal tokens" that the RL model has mastered but the base model has not.
>
> Centering: We subtract the sequence mean to maintain training stability. This ensures the resulting token weights are normalized and centered near 1, preventing the average weight from becoming too extreme.
>
> Sigmoid: The sigmoid smoothly maps the normalized loss to non-negative, bounded weights. Multiplying by 2 establishes a maximum weight near 2, keeping the overall effect within a stable range relative to the baseline (1).
>
> Clipping: This is a crucial regularization term. It restricts weights (e.g., $1 \pm 0.2$) to prevent any token from being drastically over- or under-emphasized, thus ensuring training stability and preserving the semantic coherence of the corpus.
>
> We acknowledge the need for quantitative backing. We commit to adding detailed theoretical motivation and an empirical ablation and hyperparameter study in the next version of the manuscript to solidify these design choices. Thank you for helping us refine the rigor of our paper.

---

> ### Author Response · Authors · 2025-11-26
> **Response Part 2**
>
> **Q4**: The paper should include the natural comparison baseline of standard knowledge distillation, which is currently absent.
>
> **A4:** We thank the reviewer for this excellent suggestion; we agree that a comparison against standard knowledge distillation (KD) is essential. We have conducted this experiment on OLMo-1B to evaluate the effect of KD using the RL model as the teacher.
>
> Our analysis, which will be included in the revised manuscript, reveals a critical distinction between the two approaches:
>
> **KD's Limitation (Short-Term Gains):** KD achieves strong performance during the annealing stage, confirming its ability to effectively distill knowledge and mimic the reference model in the short term. However, when we proceed to the subsequent post-training phase, the advantage gained by KD rapidly deteriorates. This suggests that KD is primarily confined to optimizing the current training loss, leading to poor generalization that fails to produce a more robust final model.
>
> **RGA's Sustainable Advantage (Transferability):** In stark contrast, RGA's performance gains achieved during the annealing stage transfer sustainably and **cascade throughout the entire post-training pipeline** (as shown in Figure 6). This demonstrates that RGA successfully integrates high-order capabilities into the foundational base model, leading to superior long-term generalization and establishing a sustainable path toward higher final model quality.
>
> - Annealing:
> | Model    | GSM8K | MATH  | GPQA  | BBH   | IFE   | HE    | MBPP  | TQA   | ARC-C | MMLU Pro | Avg.  |
> |----------|-------|-------|-------|-------|-------|-------|-------|-------|-------|----------|-------|
> | Standard | 48.14 | 10.26 | 22.54 | 30.87 | 16.19 | 8.54  | 4.60  | 22.40 | 46.67 | 13.31    | 22.35 |
> | RGA      | **61.64** | 14.50 | **24.55** | 32.07 | 28.54 | 12.80 | 9.20  | 25.58 | **49.23** | 17.44    | 27.56 |
> | KD       | 61.26 | **15.88** | 23.88 | **32.11** | **45.56** | **13.41** | **13.20** | **25.83** | **49.23** | **17.45** | **29.78** |
>
> - SFT:
> | Model    | GSM8K | MATH  | MATH-500 | MBPP  | MBPP+ | HE    | HE+   | ARC-C | GPQA Diamond | MMLU Pro | IFE   | Avg.  |
> |----------|-------|-------|----------|-------|-------|-------|-------|-------|--------------|----------|-------|-------|
> | Standard | 42.61 | 12.28 | **11.4**  | 16.2  | 25.66 | 21.95 | 18.90 | 40.96 | 24.75       | 14.34    | 56.83 | 25.99 |
> | RGA      | **47.00** | **16.54** | 10.4     | 19.6  | 29.10 | 26.22 | 21.95 | **41.38** | **25.76** | 13.56    | **59.11** | 28.24 |
> | KD       | 41.55 | 15.22 | 10.6     | **21.2** | **33.33** | **30.49** | **26.22** | 38.91 | 25.25       | **15.41** | 58.03 | **28.75** |
>
> - DPO:
> | Model    | GSM8K | MATH  | MATH-500 | MBPP  | MBPP+ | HE    | HE+   | ARC-C | GPQA Diamond | MMLU Pro | IFE   | Avg.  |
> |----------|-------|-------|----------|-------|-------|-------|-------|-------|--------------|----------|-------|-------|
> | Standard | **57.70** | 15.10 | **16.0** | 9.20  | 18.78 | 24.39 | 21.95 | 43.43 | 21.72       | 14.61    | 67.39 | 28.21 |
> | RGA      | 55.04 | **17.96** | 15.0  | 12.40 | 21.69 | **25.61** | **22.56** | **45.22** | 23.23 | 15.78 | **69.18** | **29.42** |
> | KD       | 49.66 | 15.96 | **16.0** | **12.80** | **22.75** | 22.56 | 20.73 | 39.25 | **26.34** | **15.80** | 67.87 | 28.16 |
>
> - RL:
>
> |Model|GSM8K|MATH|MATH-500|MBPP|MBPP+|HE|HE+|ARC-C|GPQA Diamond|MMLU Pro|IFE|Avg.|
> |----------|-------|-------|----------|-------|-------|-------|-------|-------|--------------|----------|-------|-------|
> |Standard|**66.72**|16.86|21.4|8.80|17.33|19.51|17.07|43.00|20.20|14.58|**70.74**|28.90|
> |RGA|65.73|**19.66**|20.2|**11.60**|**22.09**|**28.35**|**24.39**|**44.80**|**25.25**|**16.73**|70.62|**31.77**|
> |KD|62.77|16.84|**21.8**|11.50|21.30|24.39|22.56|41.38|24.75|15.24|69.30|30.17|

---

> ### Author Response · Authors · 2025-11-26
> **Response Part 3**
>
> **Q5**: Could you clarify the precise timing of the annealing phase relative to pre-training and define the "Pre-Annealed" baseline at line 306?
>
> **A5:** Thank you for this precise question, which allows us to clarify the structure of the LLM training pipeline we are operating within.
> The annealing (mid-training) stage is a classic, established stage within the overall pre-training phase, not after its completion. As visually represented in Figure 4 , the pre-training pipeline consists of Stable Pre-Training followed by the Annealing Stage. This annealing stage is identified as a critical turning point that follows the bulk of stable pre-training. Although training continues using the next-token prediction objective, this stage is characterized by two specific conditions: the introduction of high-quality corpora and the application of a rapidly decaying learning rate. The purpose is to accelerate the acquisition of higher-order capabilities and induce a substantial distributional shift in the base model. RGA operates by optimizing this critical intermediate stage.
>
> The Pre-Annealed baseline is the model checkpoint that serves as the **starting point** for the annealing phase. For our experiments, this typically refers to officially released open-source checkpoints like OLMo[1]. Specifically, it is the checkpoint that has completed the initial, long Stable Pre-Training phase. This checkpoint is obtained before the model begins any annealing or high-quality data injection, ensuring it serves as the identical starting model for all our subsequent annealing comparisons.
>
> [1] https://github.com/allenai/OLMo/tree/main
>
> **Q6**: Please clarify if the consistent downstream performance gain observed in Figure 6 is truly unique to RGA-trained models, or if the same transfer effect holds for any model that achieves strong pre-training performance.
>
> **A6:** As shown in Q4 and A4, the downstream gain is **not** guaranteed simply by achieving strong pre-training metrics.
>
> The KD-trained base model showed stronger pre-training performance than RGA. However, KD's advantage diminished rapidly in post-training, failing to produce a robust final model.
>
>
> **Q7**: In the implementation, do you put a stop gradient operator on the weight in (6)? Why or why not?
>
> **A7:** Thank you for the insightful implementation question. The weight $w_t$ is not a learnable parameter, but a scalar derived through a transformation of the loss difference between the current and RL reference models.
>
> We appreciate your insights and hope the responses address your concerns. We would be truly grateful for your reconsideration of our manuscript.

---

### Official Review · Reviewer_X8u1 · 2025-10-31

**Soundness:** 3
**Presentation:** 2
**Contribution:** 3
**Rating:** 6
**Confidence:** 3

**Summary:**

This paper introduces RL-Guided Annealing (RGA), a new LLM training method that bridges pre-training and post-training. Traditional pipelines train models through pre-training, fine-tuning, and followed by reinforcement learning. RGA proposes using the RL-tuned model to guide the mid-training during the pre-training phase (annealing stage where the learning rate decays and data quality increases). By comparing per-token losses between the base and RL models, RGA assigns token-weights, emphasizing those most influential to reasoning improvements. Experiments demonstrate consistent gain in pre-training benchmarks as well as downstream results.

**Strengths:**

* Intriguing idea using downstream training signals to influence the pre-training process, potentially closing the gap between stages that are usually isolated.
* The method is lightweight and easy to integrate. It introduces no new models or objectives, only a simple token-level reweighting mechanism that can fit neatly into existing training pipelines.

**Weaknesses:**

* Since pre-training typically targets broad, domain-agnostic language understanding, steering it with RL-trained signals might bias the model toward narrower reasoning or stylistic patterns, potentially harming generalization e.g. unseen tasks during the training including pre-training, fine-tuning and RL.
* The reported gains are relatively modest, especially given that the approach effectively requires a second pre-training phase, which can be computationally very expensive.

**Questions:**

* How sensitive is RGA to quality of the RL model used as reference. Does a weaker RL checkpoint still help, or could it misguide the annealing?
* What would happen if token reweighting were applied beyond the annealing stage, or throughout pre-training more continuously?
* How does RGA behave when the RL-tuned model is heavily domain-specific (e.g., math or code)—does it transfer well, or overfit to that domain’s distribution?

---

> ### Author Response · Authors · 2025-11-26
>
> Thank you for acknowledging the strong performance and innovative aspects of our work. Your comments have been instrumental in highlighting areas for improvement.
>
> **Q1**: What would happen if token reweighting were applied beyond the annealing stage, or throughout pre-training more continuously?
>
> **A1:** Our focus on the annealing stage is a conscious trade-off for maximal return on investment. This short, high-quality phase alone delivers a significant performance surge (e.g., a 6-point gain in average downstream performance, Table 1). Applying RGA continuously throughout the much longer stable pre-training phase would drastically increase resource consumption with potentially marginal gains.
>
> **Q2**: How sensitive is RGA to quality of the RL model used as reference?
>
> **A2:** We thank the reviewer for raising the crucial question regarding RGA's sensitivity to the reference RL model's quality and domain specificity. We agree that the reference selection is vital. We specifically choose the RL-enhanced version of the same model because it avoids the effort of selecting or training an external reference (as it is already produced by the standard pipeline) and, critically, it facilitates our primary goal of establishing a "co-improving flywheel" (Figure 4). This approach ensures RGA promotes continuous improvement throughout the entire pipeline: a better RL model leads to a better base, which facilitates further RL. We recognize the importance of this analysis and will include additional experimental studies on reference model selection in the final manuscript.
>
> Thanks again for your thorough review and valuable feedback. We hope the responses address your concerns adequately.

---

> > ### Comment · Reviewer_X8u1 · 2025-11-27
> >
> > I thank the authors for providing clarifying comments. I have also reviewed the concerns raised by other reviewers and will update my score accordingly.
> >
> > **Re: A1** I remain concerned that applying this reweighting during the entire pretraining may result in degrading pretraining performance, as the model might be overly optimized for the downstream task. I respectfully disagree with the authors' assessment that this would result in only "marginal gains"; the risk of overfitting seems non-negligible here.
> >
> > **Re: A2** I would appreciate it if the authors could provide further insight into the connection between the quality (or specific performance metrics) of the RL model and the final outcome. The current response suggests that further investigation is needed to fully understand the mechanism of the proposed algorithm.
> >
> > **Re Q3** The authors did not address my third question: "How does RGA behave when the RL-tuned model is heavily domain-specific (e.g., math or code)—does it transfer well, or overfit to that domain’s distribution?" This remains an open concern.

---

### Official Review · Reviewer_Nq79 · 2025-11-01

**Soundness:** 2
**Presentation:** 3
**Contribution:** 2
**Rating:** 2
**Confidence:** 4

**Summary:**

This work proposes RL-guided annealing (RGA) as a mid-training method aimed at better aligning the base model with the RL-tuned model, so that the annealed model becomes more receptive to later RL fine-tuning and achieves higher overall performance. The core idea is to use the RL model as a reference and re-weight tokens through importance sampling based on per-token loss differences, applied on a small set of high-quality corpora during the annealing phase. This alignment process between pre-training and post-training gives gains across several benchmarks.

**Strengths:**

The paper is generally well-written and easy to follow. The proposed RL-guided annealing method is novel and presents an interesting way to connect pre-training and post-training stages.

**Weaknesses:**

Please see Questions.

**Questions:**

1. While the method seems interesting and somewhat novel at first glance, there are a few concerns. It is not new to mid-train on data (for example, math) that the post-training procedure will later see (RL on math), in order to get a better base model for RL (as seen, for instance, in Qwen3). So it’s not entirely surprising that this setup gives some gains, which makes the extent of novelty here unclear. Moreover, Fig. 7 shows diminishing returns with successive RGA steps, raising questions about the scalability of the approach.

2. A crucial baseline seems missing. What happens if the compute spent on RGA is instead used to extend the RL phase? Also, it took me a while to realize that the “standard” baseline corresponds to mid-training on the same data without importance weighting — this could be stated more clearly.

3. It’s also not very clear under which context this method is actually useful. It feels like it might risk mode collapse, and Fig. 7 already hints at that. In practice, it’s hard to see when this would be preferable to simply spending the compute on standard mid-training or RL.

4. What is the motivation for Fig. 2? Since the annealing procedure explicitly aligns the model toward the RL model, that outcome seems almost tautological.

---

> ### Author Response · Authors · 2025-11-26
> **Response Part 1**
>
> Thank you for your insightful feedback. We appreciate the opportunity to clarify our work and are confident in its novelty and contribution.
>
> **Q1**: The method's precise novelty and practical utility require further clarification.
>
> **A1:** Thank you for this question, as it allows us to clarify a potential misunderstanding regarding our core contribution. We agree that simply mid-training on specialized data, as in Qwen3, is a known strategy.
> However, **RGA's novelty is not in the data it uses, but in how it uses that data**.
> To be explicit: our experiments use the **exact same annealing corpus** for both RGA and the baselines. The significant performance gains (e.g., 5.2% average improvement and 6x speedup on OLMo-1B) are a direct result of our **novel dynamic token-level reweighting mechanism**. This innovation leverages the existing RL-tuned model as a reference to guide the annealing phase, allowing the base model to focus on tokens critical for reasoning.
> This mechanism is also the key to RGA's practical utility. Its context is the standard LLM training pipeline, where it provides a stronger foundation before the post-training phase (Table 1). As demonstrated in our post-training results (Figure 6), this stronger base then fuels a more effective final RL model.
> In short, RGA achieves a better base model **without the costly and difficult process of curating specialized data**. We believe this practical, "self-reinforcing" method is a significant and novel contribution.
>
> **Q2**: The diminishing returns shown in Figure 7 raise scalability questions and suggest a potential risk of mode collapse.
>
> **A2:** Thank you for raising this critical point. We believe it's useful to address the two concerns—mode collapse and diminishing returns—separately.
>
> **1. Why RGA Avoids Mode Collapse:** We agree this is a critical risk, and RGA was explicitly designed to prevent it.
>
> **By Design:** Unlike hard-selection methods, RGA reweights tokens rather than discarding them, preserving full semantic coherence. Furthermore, our clipping mechanism (Eq. 5) acts as a regularizer, preventing the model from over-emphasizing any token and keeping the objective stable.
>
> **By Evidence:** The strongest evidence against mode collapse is in Table 1. RGA delivers broad performance gains across 10 diverse benchmarks. A model suffering from a collapsed distribution would fail to generalize in this way.
>
> **2. Why Diminishing Returns are Expected (and Not a Failure):** The returns diminish simply because we are iterating on the exact same, fixed dataset. **RGA is a more efficient method of extracting knowledge, but it cannot create new information beyond the 'information ceiling' of that corpus.**
>
> The purpose of the $RGA^2$ experiment was not to show infinite scalability, but to serve as a proof-of-concept for our "co-improving flywheel". It successfully shows that a better RL model (from RGA) can be 'fed back' to create an even better base.The main practical benefit comes from the highly effective first pass, which yields a superior base (Table 1) and fuels a superior final RL model (Figure 6). This is where RGA's value lies.
>
> **Q3**: What happens if the compute is spent on simply extending standard mid-training or RL？
>
> **A3:** Our core motivation is that RGA provides a **strategic optimization** that is superior to simply spending more compute time on standard training. RGA's primary goal is to establish a foundational quality improvement via the "co-improving flywheel" (Figure 4).
>
> To directly address your question, we conducted experiments on OLMo-1B where we simply extended the standard second-stage RL training. Our results show that extended training **does not always guarantee performance gains**.
>
> | Model        | GSM8K | MATH  | MATH-500 | MBPP  | MBPP+ | ARC-C | GPQA Diamond | MMLU Pro | IFE   | Avg.  |
> |--------------|-------|-------|----------|-------|-------|-------|--------------|----------|-------|-------|
> | Standard RL1 | **66.72** | 16.86 | 21.40   | 8.80  | 17.33 | 43.00 | 20.20       | 14.58    | 70.74 | 31.07 |
> | Standard RL2 | 66.03 | 19.14 | **22.20** | 10.00 | 17.86 | 44.11 | 20.20       | 14.63    | **70.86** | 31.67 |
> | RGA RL1      | 65.73 | **19.66** | 20.20   | **11.60** | **22.09** | **44.80** | **25.25** | **16.73** | 70.62 | **32.96** |

---

> ### Author Response · Authors · 2025-11-26
> **Response Part 2**
>
> **Q4**: The motivation of Figure 2.
>
> **A4:** The purpose of Figure 2 is to visually demonstrate **why the annealing phase is the critical turning point for the base model, and therefore, the ideal stage to apply RGA**.
>
> As the figure and text illustrate, we observed that the RL model's token-level probability distribution is much closer to the annealed model, but differs significantly from the pre-annealed model. This observation is key, as it **suggests that annealing plays a pivotal role in the qualitative transformation of the base model**.
>
> Based on this insight, we chose to apply RGA during the annealing phase, rather than the stable pre-training stage.
>
>
> **Q5**: The definition of the "standard" baseline could be stated more clearly.
>
> **A5:** We will add this explicit clarification to the final version of the manuscript to improve readability. Thank you for this helpful suggestion.
>
> We appreciate your insights and hope the responses address your concerns. We would be truly grateful for your reconsideration of our manuscript.

---

### Meta-Review · Area_Chair_9QFt · 2026-01-06

**Summary:**

The reviewers generally agree that the paper presents an interesting and potentially useful idea—using an RL-trained model to guide token-level reweighting during the annealing phase of pre-training—but raised substantial concerns about rigor, baselines, and cost–benefit justification.

Key concerns include: (1) limited conceptual and theoretical justification for the specific token reweighting design choices (e.g., centering, sigmoid mapping, clipping), which many reviewers found ad hoc; (2) missing or insufficiently strong baselines, particularly comparisons against simpler alternatives such as extended RL, learning-rate or curriculum adjustments, and standard knowledge distillation, making it unclear whether RGA offers unique advantages relative to its additional cost; (3) lack of clarity on when and in what practical settings rerunning annealing followed by another full post-training cycle is preferable; and (4) modest gains on larger models, raising questions about scalability and generality.

The rebuttal addressed some points with additional experiments (e.g., extended RL and KD comparisons) and provided clearer explanations of the intended pipeline and motivation. However, these responses did not fully resolve the core concerns around justification, necessity of rerunning annealing, and whether the observed gains clearly outweigh simpler or cheaper alternatives. As a result, while the idea is viewed as promising, the overall assessment remains that the evidence and analysis are not yet sufficiently comprehensive or rigorous to support acceptance.

**Reviewer Concerns:**

**Concerns largely addressed in the rebuttal**

- **Missing baselines vs. extended RL / knowledge distillation**
  (Reviewer **Nq79**, Reviewer **7oZ9**, Weaknesses / Q6):
  The authors added new experiments comparing RGA against extended RL training and standard knowledge distillation using the RL model as teacher. These results suggest that while KD or extended RL can yield short-term gains during annealing, their advantages do not consistently transfer through subsequent post-training, whereas RGA’s gains are more stable. This addresses a key baseline concern raised by multiple reviewers.

- **Clarification of the training pipeline and timing of annealing**
  (Reviewer **Nq79**, **7oZ9**, Q4):
  The rebuttal clearly explains the role of the annealing stage within pre-training and clarifies the “Pre-Annealed” baseline, resolving earlier confusion about the experimental setup.

- **Motivation for focusing on the annealing stage**
  (Reviewer **Nq79**, **A1hy**, Q2):
  The authors provided clearer empirical motivation showing that annealing is the phase where the base model’s distribution shifts most toward the RL-enhanced model, supporting the choice of applying RGA at this stage.

---

**Remaining limitations**

- **Heuristic nature of token reweighting design choices**
  (Reviewer **7oZ9**, Q3.1–Q3.4):
  While the rebuttal offers qualitative explanations for centering, sigmoid mapping, clipping, and relative loss differences, these choices remain largely heuristic. No systematic ablation or deeper analysis is provided to demonstrate necessity or robustness of these specific components.

- **Lack of comparison to simpler annealing or learning-rate baselines** *(**AC** concern)*:
  The rebuttal does not directly compare RGA against simpler alternatives such as using a more moderate learning-rate decay or higher learning-rate floor during annealing. As prior work has shown that aggressive LR decay can underutilize high-quality data, it remains unclear to what extent RGA’s gains reflect targeted token-level guidance versus an implicit reallocation of effective learning rate.

- **Cost–benefit and practical applicability concerns**
  (Reviewer **Nq79**, **X8u1**):
  Although the authors argue that annealing is relatively inexpensive and that RGA yields transferable gains, the rebuttal does not quantitatively assess whether rerunning annealing (followed by another post-training cycle) is preferable to simpler or cheaper alternatives.

- **Generalization and domain-specific bias risks**
  (Reviewer **X8u1**, follow-up):
  Concerns remain about whether guiding annealing with an RL model could bias the base model toward narrow reasoning patterns or domain-specific behaviors (e.g., math or code), as no experiments directly address this scenario.

- **Scalability to larger models**
  (Reviewer **A1hy**):
  Results on larger-scale models are not yet available, leaving open questions about whether the observed gains persist at scale.

Overall, while the rebuttal resolves several clarity and baseline-related issues, core concerns regarding justification, simplicity of alternatives, and practical tradeoffs remain.

**Reviewer Scores:**

- **Reviewer Nq79**: The rebuttal addressed several baseline and clarification concerns (extended RL, KD comparison, pipeline clarity), but core concerns about practical usefulness and cost–benefit tradeoffs remain. A modest increase from **2 → 4** seems reasonable.

- **Reviewer 7oZ9**: While the rebuttal provided qualitative explanations and added KD baselines, the reviewer’s main concerns about heuristic design choices, lack of rigorous justification, and insufficient ablations were not fully resolved. A slight improvement in clarity, but likely **2 → 4** at most.

- **Reviewer A1hy**: Concerns about limited scale, modest gains on larger models, and relation to standard distillation remain largely unchanged. The rebuttal acknowledged these limitations but did not add new large-scale evidence. Likely **4 → 4**.

- **Reviewer X8u1**: Although initially positive, the reviewer raised additional concerns after the rebuttal regarding overfitting risks, sensitivity to the RL reference quality, and lack of analysis for domain-specific RL models, explicitly noting that several questions remained unanswered. Given this follow-up, a slight downward revision from **6 → 5** appears plausible.

Overall, based on explicit post-rebuttal comments and the extent to which concerns were addressed, a reasonable post-discussion interpretation of the reviews is approximately **4, 4, 4, and 5**, corresponding to a **borderline rejection** profile.

---

### Decision · Program_Chairs · 2026-01-26

Reject